# Towards Visual Re-Identification of Fish using Fine-Grained Classification for Electronic Monitoring in Fisheries

Samitha Nuwan Thilakarathna*[1], Ercan Avsar[1], Martin Mathias Nielsen[1], and Malte Pedersen[2]

[1]DTU Aqua - National Institute of Aquatic Resources, Technical University of Denmark
[2]Visual Analysis and Perception Laboratory, Aalborg University
{msam, erca, mmani}@aqua.dtu.dk, mape@create.aau.dk

## Abstract

Accurate fisheries data are crucial for effective and sustainable marine resource management. With the recent adoption of Electronic Monitoring (EM) systems, more video data is now being collected than can be feasibly reviewed manually. This paper addresses this challenge by developing an optimized deep learning pipeline for automated fish re-identification (Re-ID) using the novel AutoFish dataset, which simulates EM systems with conveyor belts with six similarly looking fish species. We demonstrate that key Re-ID metrics (R1 and mAP@k) are substantially improved by using hard triplet mining in conjunction with a custom image transformation pipeline that includes dataset-specific normalization. By employing these strategies, we demonstrate that the Vision Transformer-based Swin-T architecture consistently outperforms the Convolutional Neural Network-based ResNet-50, achieving peak performance of 41.65% mAP@k and 90.43% Rank-1 accuracy. An in-depth analysis reveals that the primary challenge is distinguishing visually similar individuals of the same species (Intra-species errors), where viewpoint inconsistency proves significantly more detrimental than partial occlusion. The source code and documentation are available at: https://github.com/msamdk/Fish_Re_Identification.git

## 1 Introduction

The sustainable management of global fisheries hinges on the availability of accurate, comprehensive, and timely data [1, 2]. Electronic Monitoring (EM) systems, which utilize cameras on fishing vessels, have emerged as a powerful and cost-effective tool for fisheries data collection, offering an alternative to traditional methods relying on on-board observers and logbooks [3]. EM systems in fisheries are used for various objectives, including documenting catches [4, 5], ensuring legal compliance with fishing regulations, including catch limits [6], and reducing discards and bycatch [5–9]. However, the increased uptake of EM systems on-board fishing vessels has

*Corresponding Author.

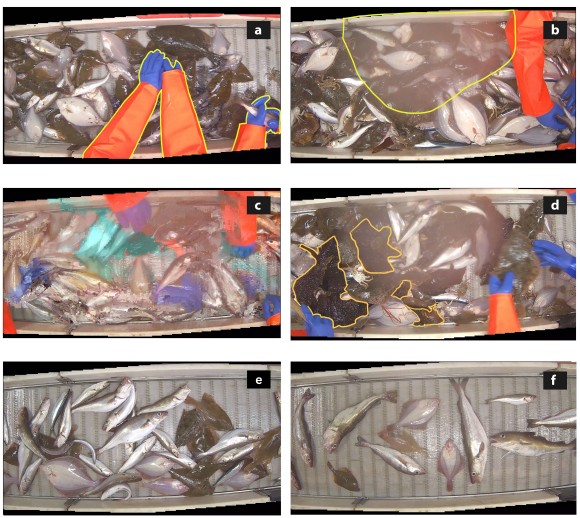

**Figure 1.** Sample images showing the diversity in image/video data encountered from EM systems. (a)-(d) show common problems: (a) obstruction of the camera view by the crew's hands; (b) lens fogging and high occlusion; (c) heavy pixelation from degraded video compression; (d) organic debris masks portions of the catch. In contrast, (e) represents a clear view of a catch with overlapping individuals, and (f) shows the ideal scenario in which the individuals are dispersed in an even layer across the conveyor belt without overlapping (Data Source: [15]).

created a new bottleneck where the volume of video data has exceeded the capacity to perform full-scale manual reviews [10, 11], a problem compounded by poor image quality, occlusions, and varied video recording conditions [8, 12–14] (Fig. 1).

Deep learning-based EM systems have emerged as a promising technology for automating catch documentation in different types of fisheries, including demersal trawling [16, 17], demersal beam trawling [18], tropical purse seines [19], and longlines [20–22]. Although these studies have shown commendable results in documenting catches in low occlusion environments, a critical challenge remains as occlusion levels increase: maintaining the identity of individual fish for accurate counting and documentation. For example, on a dynamic conveyor belt, fish are frequently occluded or move out of view, making it difficult to determine if a fish is being seen for

Proceedings of the 7th Northern Lights Deep Learning Conference (NLDL), PMLR 307, 2026.

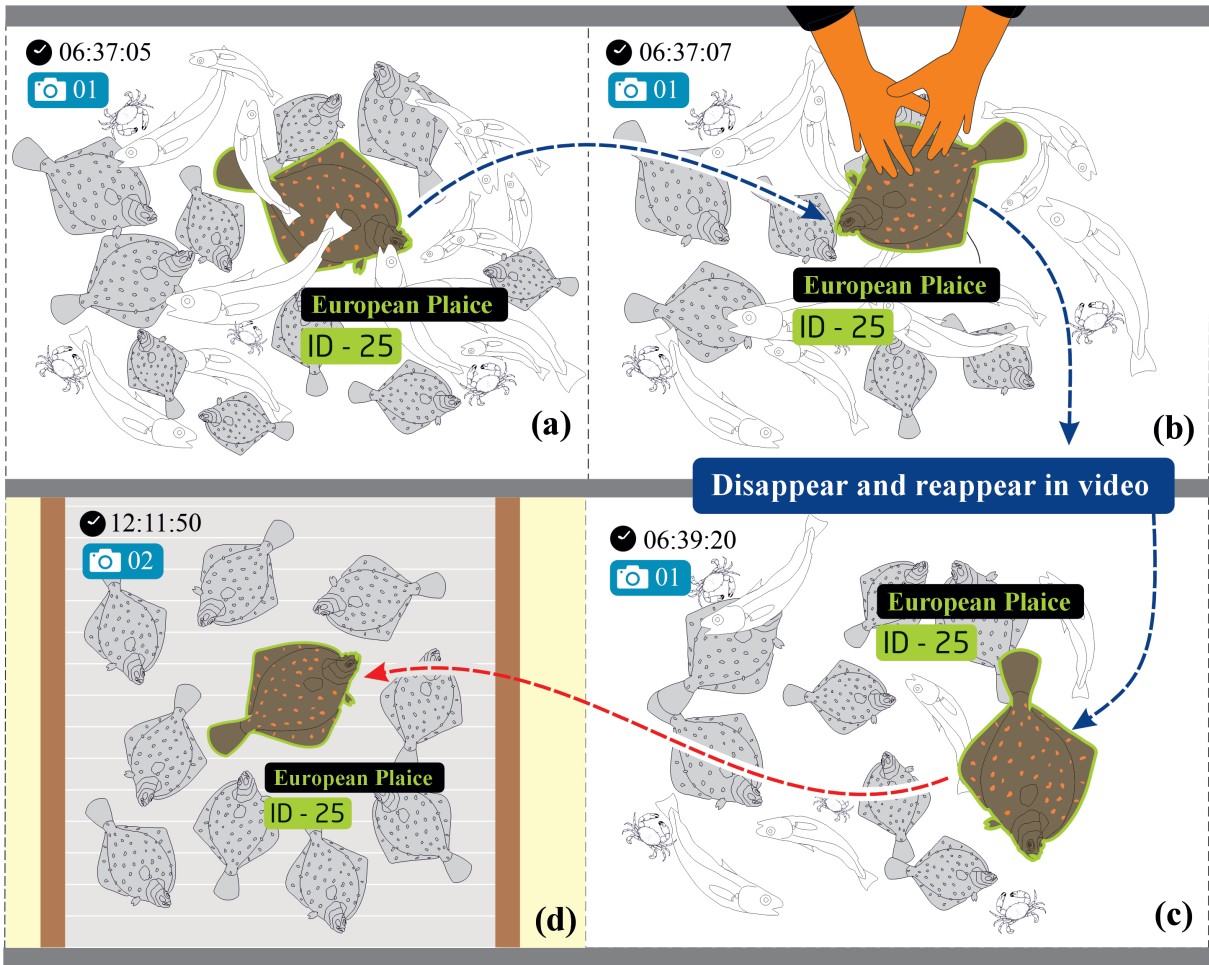

**Figure 2.** A conceptual diagram illustrating re-identification (Re-ID) applied to an Electronic monitoring system with a conveyor belt. (a) An individual is detected and assigned a unique identity (ID-25). (b-c) The system performs re-identification, successfully matching the fish after it's handled, disappears, and reappears with a new orientation within the same camera view on the conveyor belt. (d) The model's capacity for long-term, inter-camera Re-ID is shown, correctly matching ID-25 hours later in a different location under a new camera. This could be a subsampling station in the fishing vessel.

the first or second time. This frequent loss of visual contact leads to a fundamental problem of identity loss, which can cause significant errors in fisheries data. To solve this, a robust approach is needed to match an individual's identity each time it appears, a task known as re-identification (Re-ID) (Fig. 2). Re-ID has been extensively applied to persons [23] and vehicles [24]. However, the application of Re-ID in the aquatic domain remains sparse. Targeted Re-ID research in the field of aquatic sciences has primarily focused on species with distinct visual patterns [25–30].

This paper evaluates deep learning architectures for fish Re-ID in a simulated conveyor belt environment using the AutoFish dataset [31]. To rigorously isolate the feature extraction performance from upstream detection errors, this study utilizes ground-truth (GT) annotations, establishing a theoretical performance ceiling for the Re-ID component. Our primary contributions are:

(1). A comparative analysis evaluating the hierarchical Vision Transformer Swin-T (Tiny) against the Convolutional Neural Network (CNN) ResNet-50, establishing the architectural advantage of self-attention mechanisms for finegrained fish Re-ID.

(2). The identification of methodological strategies, including custom image transforms and hard triplet mining, that significantly improve Re-ID performance.

(3). An in-depth analysis of the trained Swin transformer's failure modes, revealing the core challenges in Fish Re-ID in partially occluded scenarios.

To the best of our knowledge, this paper presents the first in-depth work on the re-identification of similarly looking commercial fish species.

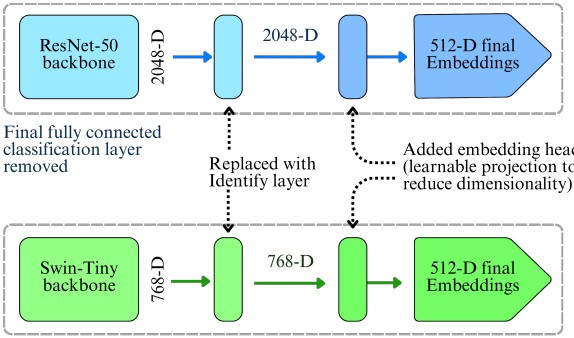

**Figure 3.** Modification of the pre-trained ResNet-50 (top) and Swin-T (bottom) backbones to produce 512-D embeddings for metric learning. In both architectures, the final classification layer is removed to expose the raw feature vector (2048-D for ResNet-50; 768-D for Swin-T). A new, learnable linear layer is then appended to each backbone to project these high-dimensional features into a final, shared 512-dimensional embedding space. D represents Dimensions.

# 2    Methodology

In the domains of person [23] and vehicle [24] Re-ID, a paradigm shift is currently underway, with Transformer-based architectures increasingly outperforming established CNN baselines. However, the transferability of these emerging architectures to the visually homogeneous aquatic domain remains underexplored.

Previous aquatic research has typically focused on single-species identification using various methodologies. Arzoumanian et al. [25], and Speed et al. [26] rely on the extraction of high-contrast keypoints (e.g., spots) to generate geometric point patterns in whale sharks.

Haurum et al [27] advanced the field by applying deep metric learning to zebrafish. While Moskvyak et al. [28] employ a landmark-guided approach that explicitly inputs annotated anatomical features (e.g., eyes, gills) to guide the learning of discriminative embeddings for manta rays, Pedersen et al. [29] utilize a keypoint matching approach for sunfish. The latter relies on detecting and matching low-level visual interest points (using descriptors like SIFT or SuperPoint) to establish geometric correspondence via homography, without requiring semantic knowledge of specific body parts. In contrast, our work addresses the more complex challenge of re-identifying individuals across multiple, visually similar commercial fish simultaneously.

In EM scenarios, relying on localized markers is precarious; if a distinguishing feature is occluded on a conveyor belt, identification fails at both the species and individual levels. To address this, we evaluate the Swin-T [32] against a ResNet-50 [33] model, under a closed-world Re-ID setting.

## 2.1    Dataset and Data preparation

The study employed the publicly available AutoFish dataset [31], a resource designed for the fine-grained analysis of fish. The dataset is composed of 1500 RGB images of 454 unique fish specimens from six most common fish species in the North Sea (Horse mackerel, whiting, haddock, cod, hake, saithe), and a miscellaneous category. Crucially for Re-ID evaluation, the dataset is pre-split into training, validation, and test sets, ensuring that there are no overlapping fish IDs between the splits and shows a similar species composition in each split. A key feature of the dataset is its structured organization designed to simulate various real-world challenges. The data is partitioned into subcategories based on two factors: **Fish arrangement and body viewpoint**.

Arrangement is categorized as either **Separated**, where fish do not overlap, or **Touched**, where fish are arranged to simulate partial occlusion. Viewpoint is categorized as either **Initial**, representing one side of the fish, or **Flipped**, representing the opposite side. This structure results in four distinct experimental conditions: Separated-Initial, Separated-Flipped, Touched-Initial, Touched-Flipped. Each fish specimen in the dataset is represented by a comprehensive set of 40 image instances, with 10 instances distributed into each of these four conditions.

For our experiments, the input data was prepared by using the GT instance segmentation masks to crop each fish, with a two pixel padding to preserve boundary details.

## 2.2    Model Architectures and Preprocessing

Our core experiment was a comparative analysis between **ResNet-50** [33], and **Swin-T** [32]. The ResNet-50 architecture, a quintessential CNN, relies on a deep stack of convolutional layers and residual connections to learn hierarchical local features. Its inductive bias is well-suited for capturing spatial hierarchies like textures and patterns. In contrast, the Swin-T architecture is a hierarchical Vision Transformer that models long-range dependencies using a shifted window self-attention mechanism. This allows it to capture global context and subtle, non-local relationships across an image. To establish a baseline, we first evaluated the off-the-shelf models in a zero-shot retrieval task, utilizing their original output dimensions (2048-D for ResNet-50, 768-D for Swin-T). Subsequently, for the fine-tuning experiments, we adapted both models by replacing their final classification layers with a unified, learnable 512-dimensional embedding head (Fig. 3).

An integral part of our methodology is a custom resize-and-pad-to-square transformation, applied to fish crops before network input. This operation first resizes each image to preserve its original aspect ratio, then pads it to fit a $224 \times 224$ canvas, ensuring that the entire fish and all local markings remain visible. In contrast, commonly used standard PyTorch training pipelines for ImageNet-style classification typically include random-sized crops or center crops after resizing, which can remove parts of the objects or heavily rescale a small region [34, 35]. Such cropping is acceptable for coarse object classification. But it is prone to discarding subtle identity cues (e.g., local patterning and fin-edge structure) that are critical for fine-grained Re-ID. Finally, we computed and applied dataset-specific normalization statistics (Channel-wise mean - [0.0495, 0.0503, 0.0535]; Standard deviation = [0.1370, 0.1363, 0.1412]) to scale the preprocessed images properly.

## 2.3 Training

Models were trained using Triplet Margin Loss (fixed margin = 0.5). To ensure each batch was effective for this loss, we used a custom PK sampler, which constructs each batch by selecting P unique fish IDs and K instances per fish ID. The batch size is the product of P and K. This guarantees the presence of positive and negative pairs in each batch. Based on the preliminary tests, we employed hard triplet mining from PyTorch Metric Learning, which selects both hard positives and hard negatives (A, $P_{\text{hard}}$, $N_{\text{hard}}$) where the negative is closer to the anchor than the positive, ensuring only challenging triplets contribute to the loss. Training ran for 300 epochs with AdamW optimizer (learning rate = $10^{-5}$, weight decay = $10^{-4}$), and a learning rate scheduler was employed to reduce the learning rate by a factor of 0.2 if the validation loss did not improve for 10 consecutive epochs.

## 2.4 Experimental Design

Our research progressed through four experimental stages. First, we established a baseline by evaluating the pre-trained models in a zero-shot retrieval task to confirm the need for fine-tuning. Second, we conducted preliminary experiments that validated our choice of custom image transforms and hard triplet mining over semi-hard triplet mining alternative by showing their superior performance. Third, the main experiment consisted of training the architectures with this optimized protocol across various batch sizes (16, 32, 64, and 256). Table 1 shows the combinations of P and K values for the selected batch sizes.

Final stage, after identifying the best-performing model, we conducted a rigorous in-depth analysis to evaluate its robustness under specific, challenging

**Table 1.** The batch size variable. P represents the number of unique fish IDs per batch and K represents the number of instances per unique fish ID per batch.

| Batch size | P | K |
|---|---|---|
| 16 | 4 | 4 |
| 32 | 4 | 8 |
| 64 | 8 | 8 |
| 256 | 32 | 8 |

conditions that simulate real-world complexities. For this, we partitioned the test set into its four subcategories as mentioned in section 2.1, and constructed distinct query and gallery sets to test the model's performance in several key scenarios systematically:

(1). **Identical Conditions**: We first established the model's baseline capabilities by performing Re-ID within identical conditions (e.g., querying Separated-Initial against a Separated-Initial gallery). This measures the model's performance when viewpoint and occlusion levels are consistent.

(2). **Viewpoint Invariance**: We tested the model's robustness to changes in viewpoint by querying fish from one side against a gallery of images showing the opposite side (e.g. Separated-Initial vs Separated-Flipped). This directly evaluates the model's ability to recognize an individual fish regardless of which side is presented to the camera.

(3). **Occlusion Robustness**: We assessed the model's resilience to occlusion by querying clearly visible fish against a gallery where the fish are touching and particularly obscured (e.g., Separated-Initial vs. Touched-Initial).

(4). **Compound Challenges**: Finally, we evaluated the model under the most difficult conditions by combining both viewpoint and occlusion changes (e.g., Separated-Initial vs. Touched-Flipped). This scenario simulates the challenging real-world task of identifying a fish that is both occluded and has been flipped over.

The detailed subcategory level experiments are listed in the Table 2.

## 2.5 Evaluation

The evaluation followed a standard Re-ID protocol where query and gallery sets were constructed programmatically from the test data. For each unique fish ID with multiple instances, one image was randomly selected to serve as the query. All remaining instances from all IDs were then used to populate

**Table 2.** Experimental setup for the in-depth subcategory analysis, detailing the query and gallery combinations used to test the model's robustness under various conditions.

| Scenario | Query | Gallery |
| --- | --- | --- |
| Identical condition scenarios | Separated-Initial | Separated-Initial |
| | Separated-flipped | Separated-flipped |
| | Touched-Initial | Touched-Initial |
| | Touched-flipped | Touched-flipped |
| Viewpoint invariance scenarios | Separated-Initial | Separated-flipped |
| | Touched-Initial | Touched-flipped |
| Occlusion robustness scenarios | Separated-initial | Touched-initial |
| | Separated-flipped | Touched-flipped |
| Compound challenge scenarios | Separated-Initial | Touched-flipped |
| | Separated-flipped | Touched-initial |

a single, comprehensive gallery. To ensure this random split is consistent and reproducible across experiments, a fixed seed was used for the selection process. The similarity between a query and each gallery image was quantified by calculating the Euclidean distance between their L2-normalized 512-dimensional embeddings, where a smaller distance indicates higher visual similarity. For each query, all gallery images were then ranked in ascending order based on this distance.

The performance of this ranking was quantified using two standard metrics. **R1 Accuracy** measures the percentage of queries where the top-ranked result is a correct identification and is defined as:

$$R1 = \frac{1}{|Q|} \sum_{q \in Q} \mathbb{I}(id(q) = id(g_{q,1})) \qquad (1)$$

Where $|Q|$ is the total number of queries, and $\mathbb{I}$ is an indicator function that is 1 if the ID of the query (q) matches the ID of its top-ranked gallery image $(g_{q,1})$.

**mean Avergae Precision (mAP) at rank k (mAP@k)** provides a more comprehensive score by evaluating the entire ranked list for each query. It is calculated by averaging the Average Precision (AP) scores across all queries in the test set $Q$. The AP for a single query q is defined as:

$$AP_{(q)} = \frac{1}{|R|} \sum_{i=1}^{n} Prec(i) \cdot Relevance(i) \qquad (2)$$

Where $q$ is a single query image being evaluated, $n$ is the total number of images in the ranked gallery list. $|R|$ is the total number of relevant images for the query $q$ that exists in the gallery (i.e., the number of other images with the same fish ID). $\sum_{i=1}^{n}$ is the summation over every position (rank) in the gallery, from the first position (i=1) to the last (i=n). $Prec(i)$ is the precision at rank i, which the proportion of correct matches found within the

top $i$ results of the ranked list. $Relevance(i)$ is an indicator function that is 1 if the image at rank $i$ is a correct match (relevant to the query) and 0 if it is not. The precision can be calculated as follows.

$$Precision = \frac{TP}{TP + FP} \qquad (3)$$

Where TP represents the number of true positives and FP represents the number of false positives. The final mAP is the mean of these AP scores:

$$mAP(Q) = \frac{\sum_{q=1}^{|Q|} AP_q}{|Q|} \qquad (4)$$

In our specific protocol, since each query has exactly 39 corresponding true matches in the gallery, we report the mAP calculated over the top 39 ranks (mAP@k = mAP@39). This provides a precise score of the model's ability to retrieve all relevant instances within those top results.

For the in-depth analysis, the same evaluation metrics were used.

## 3 Results

### 3.1 Preliminary experiments and Baseline

The preliminary experiments in Fig. 4 showed that hard triplet mining consistently outperformed semi-hard triplet mining for both Swin-T and ResNet-50. Our custom image transformation further improved R1 and mAP@k for Swin-T, and although it caused a slight R1 decrease for ResNet-50 under hard triplet mining. Its overall benefit was confirmed by higher mAP@k. Based on the clear superiority of this combination, an optimized protocol using our custom image transformation and hard triplet mining was adopted for all main experiments.

The initial zero-shot retrieval task confirmed the necessity of domain-specific training. Both models

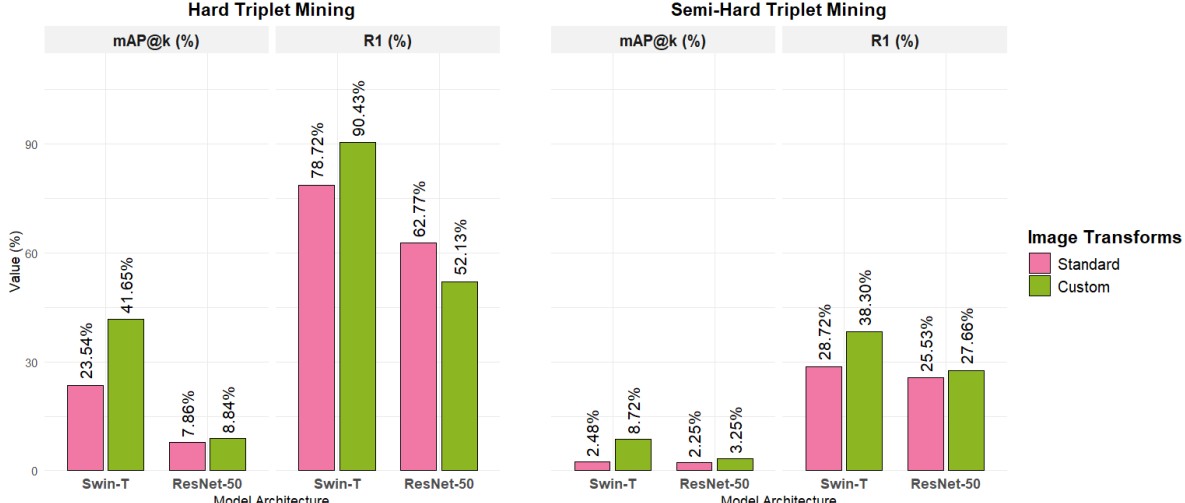

**Figure 4.** Performance comparison of Swin-T and ResNet-50 architectures utilizing standard vesus Custom image transformations. Results are stratified by triplet mining strategy (Hard vs. Semi-Hard) and evaluated on mAP@k and R1 Accuracy metrics.

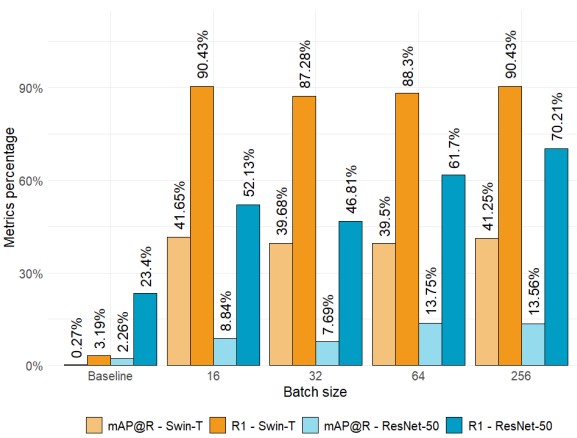

**Figure 5.** Performance trends of the fine-tuned Swin-T and ResNet-50 architectures across various batch sizes, measured by R1 accuracy and mAP@k. The Baseline group shows the initial performance of the pre-trained models for reference and the rest of the chart shows the fine-tuned performance of the model architectures.

performed poorly, with Swin-T achieving only 3.19% R1 and 0.27% mAP@k, and ResNet-50 achieving 23.40% R1 and 2.26% mAP@k (Fig. 5).

## 3.2 Model architecture comparison for Re-ID

In the main experiment comparing the fine-tuned architectures across various batch sizes, Swin-T consistently and significantly outperformed ResNet-50 (Fig. 5). The Swin-T architecture demonstrated remarkable stability and high performance across all tested batch sizes, achieving over 87% R1 and 39% mAP@k in all configurations. Peak performance

was achieved with a batch size of 16 yielding 90.43% R1 accuracy and 41.65% mAP@k. In contrast, the ResNet-50 model's performance was highly dependent on the batch size, improving as the batch size increased but never approaching the performance of Swin-T. Its peak performance was achieved with a batch size 256, reaching 70.21% R1 and 13.56% mAP@k. These results establish the clear architectural superiority of the Swin-T for this fine-grained fish Re-ID task.

To assess the learned feature space, we analyzed KDE and t-SNE plots of test set embeddings. The KDE plots (Fig. 6) show that Swin-T achieves a clear separation between positive and negative distance distributions with minimal overlap, indicating compact same-identity clusters and well-separated different identities. The ResNet-50 exhibits substantial overlap, consistent with its lower performance. The t-SNE visualizations (Fig. 7) further confirm that the Swin-T embeddings form distinct, well-separated clusters for different fish IDs, while ResNet-50 embeddings are sparsely distributed and poorly clustered with intermingled identities.

## 3.3 In-Depth Analysis

A rigorous in-depth analysis revealed a clear performance hierarchy across the various experimental scenarios (Table 3). Using the Separated-Initial subset as the primary query reference (first row), our analysis reveals a distinct performance hierarchy governed by feature correspondence (viewpoint) rather than feature completeness (occlusion). The identical condition established a near-perfect upper bound (yellow zone), confirming the model's extraction capability under ideal conditions.

**Table 3.** In-depth analysis of the best-performing model (Swin-T). The table shows the mAP@k — R1 accuracy (%) for Re-ID performance across the four dataset subcategories. Cell colors highlight specific interaction types mentioned in Table 2 (**Yellow**: Identical condition scenarios; **Red**: Viewpoint Invariance scenarios; **Blue**: Occlusion robustness scenarios; **Green**: Compound challenge scenarios).

| | Gallery | | | |
| Query | Separated Initial | Separated Flipped | Touched Initial | Touched Flipped |
|---|---|---|---|---|
| **Separated Initial** | 95.40% — 100.00% | 69.24% — 78.59% | 78.48% — 95.42% | 55.41% — 71.88% |
| **Separated Flipped** | 67.93% — 76.81% | 93.64% — 100.00% | 55.11% — 70.64% | 79.87% — 95.96% |
| **Touched Initial** | 79.70% — 89.89% | 56.94% — 63.51% | 77.59% — 100.00% | 49.45% — 59.26% |
| **Touched Flipped** | 55.29% — 62.34% | 81.33% — 89.79% | 48.31% — 58.62% | 77.64% — 100.00% |

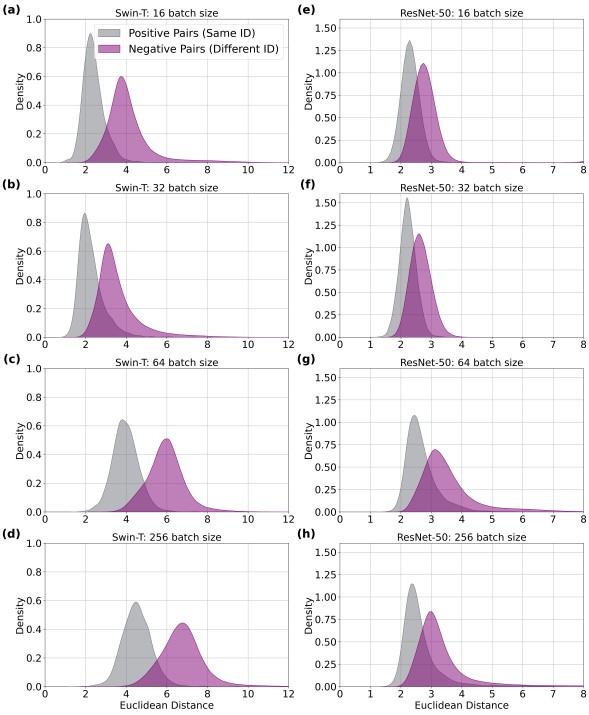

**Figure 6.** Kernel Density Plots (KDE) of the main experiments with Swin-T (a-d) and ResNet-50 (e-h). Grey density curves represent the distribution of Euclidean distances between positive pairs (same fish ID), and purple curves represent negative pairs (Different fish IDs).

**Table 4.** Summary table of rank-1 inter-species and intra-species confusions recorded in four subcategory level in the test dataset.

| Subcategory | Intra-species errors | Inter-species errors |
|---|---|---|
| Separated Initial side | 3 | 0 |
| Separated Flipped side | 5 | 0 |
| Touched Initial side | 21 | 0 |
| Touched Flipped side | 12 | 1 |

shows the lowest performance, as expected due to the combination of viewpoint as well as the occlusion. This trend holds true regardless of the query subset, reinforcing the conclusion that viewpoint consistency is the critical limiting factor for performance.

Furthermore, the Separated-Initial query achieves a slightly higher mAP@k against the Touched-Initial gallery (78.48%) than the Touched-Initial query does against itself (77.59%). This pattern recurs in the flipped scenario (79.87% vs 77.64%). This counterintuitive finding indicates that query integrity is paramount; a holistic, non-occluded query generates a more discriminative feature representation than a partially occluded one.

Critically, an analysis of the Rank-1 errors revealed that the model's mistakes were almost exclusively intra-species errors, confusing a fish with a different individual of the same species. Inter-species errors were nearly non-existent. This finding demonstrates that the Swin-T model is exceptionally robust at species-level discrimination and that the primary challenge for fine-grained fish Re-ID lies in differentiating between visually similar individuals (Table 4).

## 4 Discussion and Conclusion

This study aimed to develop and evaluate an optimized deep learning methodology for individual fish Re-ID in a controlled setting simulating an EM context. Our findings demonstrate the superiority of the Swin Transformer architecture and highlight critical methodological choices required for this fine-

As we introduce challenges, a clear divergence emerges: the model demonstrates remarkable performance to partial occlusion in the Touched-Initial gallery scenario (blue zone), maintaining a R1 accuracy of 95.42% and a mAP@k of 78.48%. This high performance is attributed to the preservation of the lateral viewpoint, allowing the model to exploit high-frequency texture details despite occlusion. In sharp contrast, the Separated-Flipped scenario (red zone)–which presents the full, non-occluded fish but from the opposite side–causes a significant drop to 78.59% R1 accuracy and 69.24% mAP@k. This confirms that the loss of asymmetric lateral features is far more detrimental to identification than partial occlusion. The compound challenge (green zone)

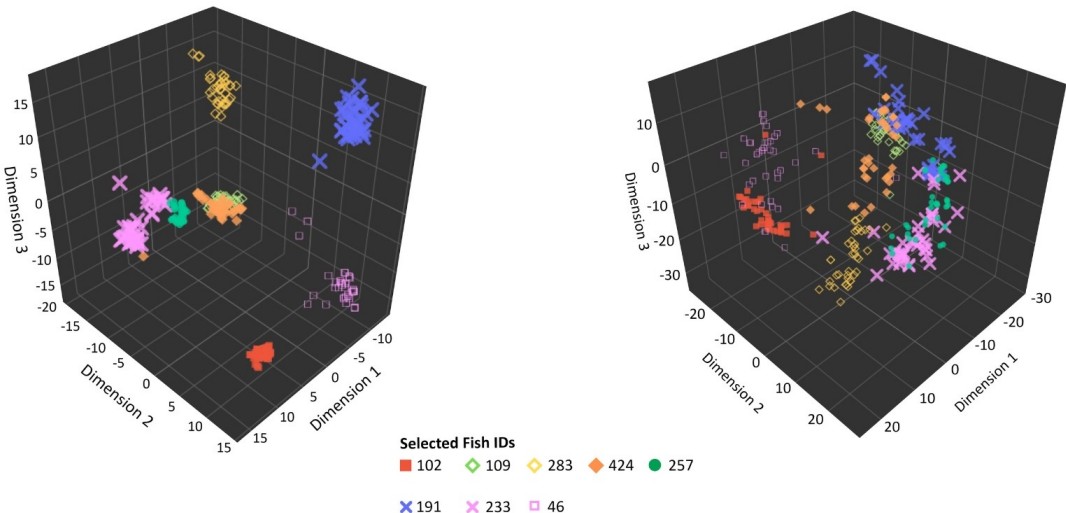

**Figure 7.** t-SNE (t-Distributed Stochastic Neighbour Embedding) plots for the best performer Swin-T (16 batch size) (Top) and the best ResNet-50 (256 batch) (Bottom). These are the gallery embedding vectors of the test dataset. Only 8 fish IDs out of 94 fish IDs in the test set are visualized in here for a clear view.

grained task. The consistent superiority of Swin-T over ResNet-50 is attributed to two key architectural differences. First, the transformer's shifted window self-attention mechanism allows the model to propagate information across the entire image, capturing global context and subtle, distributed biological markers that local CNN receptive fields may miss. Second, our experiments revealed that Swin-T maintains high performance even at smaller batch sizes, whereas ResNet-50 performance degrades. We attribute this to the difference in normalization strategies: ResNet relies on batch normalization (BN), where the calculation of mean and variance becomes unstable with small batches [36]. In contrast, Swin-T employs layer normalization (LN), which computes statistics independently for each sample [37]. This makes the transformer architecture significantly more robust for tasks where hardware constraints limit batch size.

Methodologically, the resize and pad to square image transformation proved essential for preserving extremity features, and a fixed hard triplet margin (m = 0.5) successfully enforced rigorous identity separation in this controlled setting. However, we acknowledge that this hyperparameter involves a trade-off. While effective for clean laboratory data, a high fixed margin may hinder convergence in real-world fishing environments characterized by high visual noise (e.g., blood, debris). Future work should prioritize an ablation study to optimize margins for noisy domains or explore adaptive margin strategies.

Crucially, our analysis reveals that viewpoint consistency is a stronger determinant of Re-ID success than feature completeness for this specific study. This confirms that the lateral viewpoint contains the primary identity signature; preserving it allows

for robust matching even under occlusion, whereas flipping the fish removes access to critical or subtle asymmetric features. Furthermore, we observe that consistency in occlusion state is secondary to query integrity. Retrieval performance decreases when using an occluded query (Touched) even against a similarly occluded gallery. In contrast, a feature-rich, non-occluded query (Separated) successfully retrieves targets even in occluded scenarios. This suggests that a complete visual signature is more valuable to the model than matching the occlusion levels between the query and the gallery.

The error analysis reveals that mistakes were almost exclusively intra-species. This indicates that future data collection efforts should prioritize acquiring Hard-Negative examples–visually similar individuals of the same species under varied viewpoints–to further refine the model's capability for fine-grained discrimination. These insights are highly valuable for developing robust automated catch recognition systems. By prioritizing high-quality query acquisition or best shot selection systems can prevent identity loss even in cluttered environments. This capability ensures reliable re-identification, directly improving the quality of catch data required for downstream tasks such as stock assessments, selectivity studies, and gear development.

# Acknowledgements

This work was funded by the European Union Horizon Europe under the OptiFish project (Grant agreement No 101136674). Computational resources and services were provided by the DTU Computing Center (DCC) [38].

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
