# OpenReview forum: "Towards Visual Re-Identification of Fish using Fine-Grained Classification for Electronic Monitoring in Fisheries"
_NLDL.org/2026/Conference — NLDL 2026 Poster_

### Official Review · Reviewer_ggpc · 2025-10-06
**Well-executed paper, with has strong potential impact. It needs minor revision.**

**Rating:** 5
**Confidence:** 3

**Summary:**

This paper addresses the challenge of automating fish monitoring in fisheries using re-identification (Re-ID) techniques. In other words, the goal is to recognise the same fish (not only the species, but also the individual) across multiple video frames.

The authors use a novel dataset (the AutoFish dataset, 2025), comparing two machine learning models: a convolutional neural network (ResNet-50) and a transformer-based model (Swin-T).

They propose tailored preprocessing methods and a training strategy (“hard triplet mining”) targeted to significantly improve performance.

Results are "simple" but impactful and linked to reality: first of all, the Swin-T architecture outperforms the CNN consistently. Secondly, the analysis shows that errors mainly occur when distinguishing between different individuals of the same species, rather than between different species.

**Strengths:**

Several strong points have been identified:

1) Relevance: The paper addresses a real-world problem (monitoring fisheries for sustainability and regulation), and it is directly applicable to the growing use of electronic monitoring systems within industry. Both the problem and how the paper wants to tackle it are clear.

2) Clarity of the technical goals: The research questions inspiring the whole investigation are clear. Different types of models/architectures are compared, the effect of training strategies is investigated, and challenges and weaknesses are clearly identified.

3) The work uses a publicly available dataset (AutoFish), which seems to be well-structured and realistic.

4) Careful experimental setup with clear conditions (viewpoint, occlusion, combined challenges).

5) Provides both quantitative results (accuracy metrics) and qualitative analyses (plots and error analysis).

6) The paper is well-written, with a logical structure, clear figures and tables. Technical terms are explained reasonably well for non-specialists most of the time.

Finally, it is worth mentioning the "in-depth analysis" section: nice touch!

**Weaknesses:**

A few weaknesses were identified:

1) The paper claims to be the first in-depth work on fish Re-ID for similar-looking species. Nevertheless, the distinction from related studies (animal/fish recognition, species classification) could be explained more clearly.

2) Some sections (e.g., triplet mining, embedding spaces) are described with heavy technical language. More intuitive explanations and some analogies could be included in order to make the paper more accessible to a broader audience.

3) The limitation that performance on real-world fisheries data could be much lower is mentioned, but it deserves stronger emphasis because its current applicability could be overestimated.

4) Elaborate on the reasons why the two models (ResNet-50 and Swin-T) were chosen. The inclusion of additional baselines or state-of-the-art methods could strengthen the analysis: consider it for a new publication (not for this one).

5) Metrics are well defined, but there is little discussion of their practical meaning (e.g., What are the stakeholders' user requirements? What does 90% Rank-1 accuracy mean in operational terms for fisheries? Is it enough, a lot or too little?).

6) In line with 5), even if the results are solid, the discussion could go deeper into the implications for fisheries management and policy.

**Justification:**

The paper demonstrates solid methodology and results, and the application is potentially impactful for fisheries monitoring (even if more discussion on this aspect is required before acceptance) and valuable to the ML community.

I recommend acceptance after minor updates.

---

> ### Author Rebuttal · Authors · 2025-10-22
>
> Dear Reviewer,
> Thank you very much for your thoughtful review and insightful comments on our manuscript. We sincerely appreciate the time you took to provide this valuable feedback.
> As this rebuttal stage does not permit manuscript revisions, our response below focuses on clarifying our methods, analysis, and rationale to address the concerns you have raised. We hope these explanations demonstrate that the points of confusion or perceived weaknesses are, in fact, addressed or justified within the current scope and structure of the paper.
> We have carefully considered each of your points and provide our detailed, point-by-point explanations below.
>
> **Weaknesses**
> 1. **The paper claims to be the first in-depth work on fish Re-ID for similar-looking species. Nevertheless, the distinction from related studies (animal/fish recognition, species classification) could be explained more clearly.**  We appreciate the reviewer’s suggestion to clarify the novelty of our work concerning existing fish recognition literature. While prior studies have addressed species-level identification and single species fish re-identification, our work establishes the first in-depth study on individual level fish Re-ID for visually similar species within the challenging context of commercial fisheries Electronic Monitoring (EM) systems. In fisheries context, the conditions are different from underwater re-identification, and we need to identify the same individual in different times to prevent miscounting the same fish. In the revised manuscript we will amend the relevant section to explicitly detail the difference in problem scope between our specific Re-ID task and the previous studies, thereby strengthening the clarity of our unique contribution.
> 2. **Some sections (e.g., triplet mining, embedding spaces) are described with heavy technical language. More intuitive explanations and some analogies could be included in order to make the paper more accessible to a broader audience.** We appreciate the suggestion of giving it more intuitive explanations rather than the heavy technical details. But the re-identification is a less explored area of deep learning, especially for the fisheries context, we thought it would be more beneficial to the fisheries community to have an interest to the application of computer vision to the fisheries monitoring as an emerging area of research. We addressed our research with minimum possible technical facts to give the clear picture of our study. But If the paper is accepted, we will try to explain the heavy technical facts with simple analogies If the page limit is conserved.
> 3. **The limitation that performance on real-world fisheries data could be much lower is mentioned, but it deserves stronger emphasis because its current applicability could be overestimated.** We agree with the reviewer. The use of this model in the real-world fisheries environment will produce lower results, because it consists of different species other than the species which the model has been trained on. So, the approach will be an open-world re-identification scenario with high level of occlusion. We would like to take this as the foundational study to investigate the feasibility of using well established deep learning model architectures as feature extractors for the fish Re-ID in commercial fisheries aspect using a controlled dataset. But we will revise the discussion section to emphasize the practical constraint of our model and the methodology.
> 4. **Elaborate on the reasons why the two models (ResNet-50 and Swin-T) were chosen. The inclusion of additional baselines or state-of-the-art methods could strengthen the analysis: consider it for a new publication (not for this one).**
> The choice to compare ResNet-50 and Swin-T was deliberate as they represent the leading and most relevant paradigm in visual recognition and Re-identification over the past years. ResNet-50 was selected because it is one of the established, highly optimized CNN based backbone used in large number of Re-ID literature, making it a good candidate for comparison. Swin-T on the other hand, was chosen because it represents one of the latest advancements in vision transformers, leveraging hierarchical feature extraction with shifted windows for efficient and scalable modelling. Our primary goal was to conduct an in-depth, direct comparison between the two dominant architectural families in the context of this novel application in fisheries. We agree that incorporating additional baselines, or other state-of-the-art specialized Re-ID networks, would further strengthen the analysis. We will reserve this valuable suggestion for future work and a subsequent publication.
> 5. **Metrics are well defined, but there is little discussion of their practical meaning (e.g., What are the stakeholders' user requirements? What does 90% Rank-1 accuracy mean in operational terms for fisheries? Is it enough, a lot or too little?).**
> We appreciate the reviewer’s observation that translating our technical metrics into practical operational terms is crucial for the fisheries community audience. We agree that this link is vital for demonstrating the system’s utility to stakeholders whose primary concern is accurate catch monitoring and regulatory compliance.
> From a political control and enforcement perspective little emphasis has been put into answering the question, “when is good, good enough?”. We acknowledge this as a crucial question to address to help steer towards implementation of automated catch documentation. Does it need to perform as accurate is human video reviewers or does the ability to review significantly larger pools of data at less accuracy suffice, and at what accuracy.
> In the present study, we take Rank-1 accuracy as a metric to demonstrate the high ability of Re-ID in reducing miscounting. For an example, the achieved Rank-1 accuracy (e.g., 90.43% for Swin-T) directly correlates to the system correctly identifies the same individual fish on its very first attempt for 9 out of every 10 retrieval queries. The results, even when derived from a controlled dataset, strongly indicate the power of Re-ID to support the accurate tracking of unique individuals, establishing its viability for integration into AI-enhanced Electronic Monitoring (EM) systems to significantly boost video review efficiency and data integrity. While this study successfully validates the core Re-ID mechanism, subsequent research will focus on developing complete, robust automated pipeline capable of handling the complexities of real-world operational environments.
> 6. **In line with 5), even if the results are solid, the discussion could go deeper into the implications for fisheries management and policy**
> Based on the findings in the present study, Re-ID appears as a promising method for obtaining high accuracy in automated catch documentation, also in occluded catch scenarios which have otherwise proven challenging. We acknowledge that a small paragraph can be added on the importance and implications of accurate data than what is currently already stated in the introduction.

---

### Official Review · Reviewer_vvrf · 2025-10-07
**Transformer-Based Fish Re-Identification for Electronic Monitoring: Strong Empirical Results, Limited Novelty**

**Rating:** 4
**Confidence:** 4

**Summary:**

This paper presents a deep-learning pipeline for fine-grained fish re-identification (Re-ID) in a simulated Electronic Monitoring (EM) setup, using the novel AutoFish dataset. The dataset simulates conveyor-belt imagery containing six visually similar fish species under controlled conditions. Each fish instance is cropped using ground-truth segmentation masks to isolate individuals, and no detection or segmentation steps are involved.

The study compares Swin Transformer (Swin-T) with ResNet-50 under a metric-learning framework using triplet margin loss. Both networks produce 512-dimensional embeddings, and similarity is measured using Euclidean distance. A key methodological contribution is the integration of hard triplet mining and a dataset-specific image normalization pipeline, which together significantly improve retrieval performance. Experiments are structured around two orthogonal conditions — Arrangement (Separated vs. Touched) and Viewpoint (Initial vs. Flipped) — to simulate realistic EM scenarios. Results demonstrate that Swin-T outperforms the CNN baseline across all tested configurations.

Overall, the paper provides a sound and reproducible benchmark for vision-based fish Re-ID, highlighting the promise of transformer-based architectures for fine-grained recognition in fisheries monitoring.

**Strengths:**

Solid empirical performance: The Swin-T model convincingly outperforms ResNet-50 under identical conditions, suggesting real architectural advantages for fine-grained feature discrimination.

Well-motivated problem: The need for automated Re-ID in EM fisheries is clearly justified and highly relevant for real-world sustainability monitoring.

Controlled dataset design: The AutoFish dataset provides a clean, interpretable testbed for evaluating appearance-based fish Re-ID without the confounding effects of detection or tracking.

Good methodological rigor: The use of hard triplet mining and dataset-specific normalization reflects thoughtful adaptation of metric learning principles to the domain.

Insightful analysis: Visualization of the learned embedding spaces (KDE, t-SNE) adds further interpretability to the results.

Reproducibility: The methods, hyperparameters, and dataset splits are described with reasonable clarity, suggesting that results could be reproduced by others.

Questions for the Authors

 - Were models trained across all arrangement/viewpoint categories jointly, or separately per condition?
 - Were embedding vectors L2-normalized before computing Euclidean distances?
 - Was the triplet loss margin ($m = 0.5$) fixed or tuned, and were alternative margins explored?
 - Why was occlusion not used as a data augmentation, given that occlusion drives the main failure cases?
 - Can the authors explain why Swin-T’s performance is less sensitive to batch size than ResNet-50?

These clarifications would strengthen confidence in the soundness and generality of the conclusions.

**Weaknesses:**

Limited novelty: The core contributions (use of Swin-T and hard triplet mining) are adaptations of well-established techniques rather than conceptual innovations. The work is best seen as an application study with careful tuning rather than a methodological breakthrough.

Ambiguity in experimental design: It is not clearly stated whether training occurred across or within the viewpoint/arrangement splits. This matters greatly for understanding generalization.

Metric details under-specified: The embedding normalization, distance computation, and margin tuning are not fully described, making the reported numbers less interpretable.

Unexplored augmentations: Occlusion is acknowledged as the main failure mode but was not simulated in training — a missed opportunity.

Interpretation overstatement: The claim that intra-species confusion is “the crucial insight” is somewhat overstated — it is an expected finding given the task definition and dataset design.

**Justification:**

This paper presents a competent and well-executed application of state-of-the-art deep metric learning to an important domain. While it lacks strong algorithmic novelty, it compensates with methodological rigor and empirical strength. The transformer-based architecture’s consistent advantage over CNNs is a meaningful and well-supported result. The paper would benefit from clearer methodological exposition and minor experimental extensions (especially regarding occlusion and margin tuning), but these do not undermine its core validity.

Verdict: Weak Accept — The work is sound, results are convincing, and the contribution is practically valuable for the EM and fisheries monitoring community, even if conceptually incremental.

An LLM was used to help turn my notes into a readable review. The submission itself was not provided to the model, and the evaluations and conclusions are entirely my own.

---

> ### Author Rebuttal · Authors · 2025-10-22
>
> Dear Reviewer,
> Thank you very much for your thoughtful review and insightful comments on our manuscript. We sincerely appreciate the time you took to provide this valuable feedback.
> As this rebuttal stage does not permit manuscript revisions, our response below focuses on clarifying our methods, analysis, and rationale to address the concerns you have raised. We hope these explanations demonstrate that the points of confusion or perceived weaknesses are, in fact, addressed or justified within the current scope and structure of the paper.
> We have carefully considered each of your points and provide our detailed, point-by-point explanations below.\
> **Answers to the questions**
> 1. **Were models trained across all arrangement/viewpoint categories jointly, or separately per condition?** The models were trained all categories jointly which include all the variations (viewpoint and occlusion configurations). We agree to include this information in the updated version of the manuscript.
> 2. **Were  embedding vectors L2-normalized before computing Euclidean distances?**
> Yes, the embedding vectors were L2 normalized, and we will update this fact in the manuscript.
> 3. **Was the triplet loss margin () fixed or tuned, and were alternative margins explored?**
> The triplet loss margin was fixed in 0.5.  The selection of this relatively higher margin is a direct consequence of the low inter-class variance inherent in the fish Re-ID problem. Due to the extreme similarity between individual fish (even the species), the initial separation of the Anchor-Positive (A, P) pair and Anchor-Negative (A, N) tend to be very small. If we used a lower value, the resulting clusters would only have a minimal buffer between them. In a fine-grained task, this minimal gap is highly vulnerable to noise and small feature variations, meaning a small change in lighting or orientation could cause a negative example to incorrectly cross into the positive cluster, leading to high false positive rates during retrieval. By setting the margin to a higher value, we intended to force the network to perform extra work to establish a significantly larger, more robust safety margin of separation. This ensures that the final feature space is much more discriminative. The distinct clusters for each individual fish are separated by a wide, unambiguous gap, making the Re-ID system strong even when presented with subtle differences in new images. We would like to add this detail into the manuscript in the revised version.
> 4. **Why was occlusion not used as a data augmentation, given that occlusion drives the main failure cases?**
> In the AutoFish dataset, the occlusion configuration is defined as “Touched”. Therefore, the occlusion is already included in the training set. The AutoFish dataset has been carefully organized to represent the separated (non-occluded) and touched (occluded) configurations for each fish ID. Therefore, we did not augment the dataset to simulate occlusion.
> 5. **Can the authors explain why Swin-T’s performance is less sensitive to batch size than ResNet-50?**
> Swin-T’s performance exhibits less sensitivity to batch size than ResNet-50 due to fundamental differences in their normalization techniques. ResNet-50 heavily relies on Batch normalization (BN). (Wu & He, 2018) demonstrated that BN’s reliance on calculating statistics (mean/variance) across the mini batch leads to noisy and inaccurate statistics when the batch size is small. This statistical instability introduces high variance into the gradients, resulting in the observed degradation in ResNet-50s performance. In contrast, Swin-T’s architecture uses layer normalization (LN) (Aburass & Dorgham, 2023). LN calculates its statistics per-sample (across the feature dimension) rather than across the batch, making its feature representation inherently dependent of the batch size. Therefore, Swin-T maintains stable training and robust feature quality even with the small batch sizes necessitated by our experimental setup. We agree to briefly mention this in the updated version of the manuscript.
>
> **Weaknesses**
> 1. **Limited novelty: The core contributions (use of Swin-T and hard triplet mining) are adaptations of well-established techniques rather than conceptual innovations. The work is best seen as an application study with careful tuning rather than a methodological breakthrough.** The core goal of the paper is an application study with the available foundational models. We evaluated two of the most established deep learning model architectures to investigate the suitability of them the Re-ID task which will be used in the overall pipeline of automated catch documentation in fisheries.
> 2. **Ambiguity  in experimental design: It is not clearly stated whether training occurred across or within the viewpoint/arrangement splits. This matters greatly for understanding generalization.** The AutoFish dataset (Bengtson et al., 2025) is designed in a way that all training, validation and test splits have both arrangement (Touched, separated) as well as viewpoint variation. But each split consists of different set of fish IDs (Individuals) which prevents the mixing of fish between the splits. These split arrangements are defined in the AutoFish paper, and we followed the exact split arrangement during our experiment. We agree to mention this fact in the manuscript before being published .
> 3. **Metric details under-specified: The embedding normalization, distance computation, and margin tuning are not fully described, making the reported numbers less interpretable.** We agree that we did not specify expressions for distance computation because it is the standard Euclidean distance. The embeddings were L2 normalized and the margin was not tuned and used a fixed margin of 0.5. Selection of the margin has already been explained in the above question, and we plan to address these points as much as we can within the page limits.
> 4. **Unexplored augmentations: Occlusion is acknowledged as the main failure mode but was not simulated in training — a missed opportunity.** As I explained in the question above, the occlusion (touched configuration) was included in the training set as well. The training, validation and test splits are all equipped with occlusion (arrangement variations) and the viewpoint variations. We did not employ a separate augmentation to simulate the occlusion.
> 5. **Interpretation overstatement: The claim that intra-species confusion is “the crucial insight” is somewhat overstated — it is an expected finding given the task definition and dataset design.** We would not agree with the idea of overstatement. This is just stating the capabilities of the deep learning model for the re-identification. This intra-species confusion is indeed expected to be addressed in the task definition. But with the use of the ground truth data from the AutoFish dataset, we gained insight into the amount of intra-species errors that are produced from the trained model. So, we can optimize the performance to reduce this error by collecting more structured data to train the models to identify extremely subtle features of fish. Besides, it has just been mentioned in a small part of the manuscript (Section 4, Lines 411-414) as a finding. In other words, the discussion part has not been dominated by this statement.
>
> **Bibliography**
> 1. Aburass, S., & Dorgham, O. (2023). Performance Evaluation of Swin Vision Transformer Model Using Gradient Accumulation Optimization Technique. In K. Arai (Ed.), Proceedings of the Future Technologies Conference (FTC) 2023, Volume 4 (pp. 56–64). Springer Nature Switzerland.
> 2. Bengtson, S. H., Lehotský, D., Ismiroglou, V., Madsen, N., Moeslund, T. B., & Pedersen, M. (2025). AutoFish: Dataset and Benchmark for Fine-grained Analysis of Fish. http://arxiv.org/abs/2501.03767
> 3. Wu, Y., & He, K. (2018). Group Normalization. http://arxiv.org/abs/1803.08494

---

### Official Review · Reviewer_tZ22 · 2025-10-08
**Evaluation of deep learning architectures for ReID of similarly looking fish species**

**Rating:** 2
**Confidence:** 4
**Final Rating:** 4
**Final Confidence:** 4

**Summary:**

This paper addresses the task of fish re-identification to automatically review data collected by Electronic Monitoring (EM) systems, an important step toward automatic catch documentation. The authors compare a CNN-based approach (ResNet-50) with a Swin Transformer–based approach in a simulated conveyor belt environment using the AutoFish dataset. In both cases, training was performed using triplet margin loss. They also compare hard and semi-hard mining strategies and incorporate a custom transformation.

**Strengths:**

+ The paper is well written and easy to read.
+ The authors tested fish re-identification on a dataset simulating a dynamic conveyor belt, where data can be occluded and some individuals can temporarily move out of view.
+ The authors conducted an analysis and used t-SNE and KDE for visualization.
+ The paper addresses the re-identification of similarly looking individuals, which is a challenging and relevant problem.

**Weaknesses:**

- The paper lacks a detailed presentation of existing ReID work and does not clearly situate the authors’ contribution within the state of the art.
- ReID has been extensively applied to persons and vehicles, but the authors did not compare their method to those approaches applied to fish.
- It would be valuable to apply or at least discuss existing fish tracking methods, for example: Belmouhcine, Abdelbadie, et al., "Fully Deep Simple Online Real-time Tracking: Efficient Re-Identification by Attention without Explicit Similarity Learning," ICPR 2022, which can also be used for ReID.
- The study assumes 100% accurate fish detection, as they use ground-truth annotations to crop fish. In a real scenario, automatic ReID would rely on automatic detection, which may suffer from false positives and false negatives.
- The paper does not specify whether hard positive mining, hard negative mining, or both were used.
- For the baseline, it seems no fine-tuning was done. But how did the authors train the learnable projection for dimensionality reduction (embedding head)?
- The default image transformation is not clearly described.
- The authors tested only Euclidean distance for retrieval; evaluating other distance metrics or similarity measures would be useful.
- In Table 3, the authors did not explain why Separate Flipped (Q) / Touched Initial (G) gives better results than Touched Initial (Q) / Touched Initial (G), and similarly for the other combinations (Separate Initial (Q) / Touched Flipped (G) and Touched Flipped  (Q) / Touched Flipped (G)).
- In the t-SNE plots, the authors did not explain why only 8 fish IDs were shown, nor how these IDs were selected.

**Final Justification:**

The authors have carefully considered most of my comments and plan to revise the manuscript accordingly. The proposed changes and suggestions strengthen the paper, and it is clear that the authors will thoughtfully incorporate the feedback received during the review process in the revised version. Although the paper lacks novelty, I consider it borderline but leaning toward acceptance after the suggested revisions, as the application is interesting and the methodology and correctness are fair enough.

**Justification:**

The paper does not contain a sufficient technical contribution. It applies existing methods to the AutoFish dataset without introducing novel techniques. More extensive experiments are needed, and the paper should compare its approach to other existing ReID methods applied to fish.

---

> ### Author Rebuttal · Authors · 2025-10-22
>
> Dear Reviewer,
> Thank you very much for your thoughtful review and insightful comments on our manuscript. We sincerely appreciate the time you took to provide this valuable feedback.
> As this rebuttal stage does not permit manuscript revisions, our response below focuses on clarifying our methods, analysis, and rationale to address the concerns you have raised. We hope these explanations demonstrate that the points of confusion or perceived weaknesses are, in fact, addressed or justified within the current scope and structure of the paper.
> We have carefully considered each of your points and provide our detailed, point-by-point explanations below.
>
> 1. **The paper lacks a detailed presentation of existing Re-ID work and does not clearly situate the authors' contribution within the state of the art**
> We agree that situating our work within the SOTA is critical to highlight its novelty. Our paper addresses a unique and highly specialized challenge: developing automated Re-ID for high-throughput commercial fishing Electronic Monitoring (EM). This required explicit differentiation from general Re-ID work to clearly define our technical contribution. Our research is situated at the intersection of Deep Metric Learning Re-ID and Electronic Monitoring Systems for Fully Documented Fisheries (FDF) compliance.
> We briefly cited the existing Re-ID works of persons, vehicles and sparsely done fish re-id studies. But we did not mention the technical details of them because of the page limit of the paper is highly restricted. In the manuscript, the following sentence cites the existing works on fish Re-ID. We agree to add an elaborated paragraph summarizing the methodological applications in the previous fish Re-ID studies (Literature review), if the paper is accepted.
>
> 2. **Re-ID has been extensively applied to persons and vehicles, but the authors did not compare their method to those approaches applied to fish**
> As mentioned in the previous response, we would like to add a methodological review section addressing the Re-ID which have been applied to the fish. We aim to elaborate on how our approach differs from the existing studies in that section.
> 3. **It would be valuable to apply or at least discuss existing fish tracking methods, for example : Belmouhcine, Abdelbadie, et al., "Fully Deep Simple Online Real-time Tracking: Efficient Re-Identification by Attention without Explicit Similarity Learning," ICPR 2022, which can also be used for Re-ID.**
> This study’s primary objective is to serve as a foundational feasibility analysis to isolate and thoroughly test the most challenging component of the overall EM solution, establishing fine-grained individual identity (Re-ID) among visually similar commercial fish. The AutoFish dataset was intentionally constructed with static, highly organized images, effectively decoupling the identity problem from the temporal tracking problem to ensure model performance relies purely on learning robust, pose invariant embeddings (Bengtson et al., “AutoFish: Dataset and Benchmark for Fine-grained Analysis of Fish”). Therefore, by limiting our scope to Re-ID, this paper establishes a clean, empirical SOTA benchmark for the identity-matching component, which is the fundamental prerequisite for any complete commercial tracking system.
> 4. **The study assumes 100% accurate fish detection, as they use ground-truth annotations to crop fish. In a real scenario, automatic Re-ID would rely on automatic detection, which may suffer from false positives and false negatives.**
> We agree that an automatic detector is necessary for real-world deployment. However, this foundational study was deliberately scoped as a proof-of-concept to isolate and benchmark the Re-ID component. By using ground-truth crops, we eliminated detection error as a confounding variable. This allows us to establish a clean, maximum attainable SOTA performance for the feature extraction capabilities (Swin-T vs. ResNet-50) alone. Given this is a novel application with no baselines, establishing this performance ceiling is a crucial first contribution before tackling the compounding errors of a full detection-and-Re-ID pipeline. We will clarify this scope and explicilty state that integrating an object detector is future work.
> 5. **The paper does not specify whether hard positive mining, hard negative mining, or both were used.**
> We apologize for the ambiguity in the original text. We confirm that our strategy utilized a strict form of hard triplet mining, which explicitly selects both hard positive and hard negative samples simultaneously. Using the TripletMarginMiner from PyTorch metric learning library, we set the configuration, “type of triplets=hard”. This configuration selects triplets that represent the most difficult cases for our model (A, Phard, Nhard). It combines the hard positive with the hard negative. Crucially, the miner only passes the triplets to the loss function if the negative is closer to the anchor than the positive. We will revise these technical details in the manuscript to make it clearer for the readers if the paper is accepted.
> 6. **For the baseline, it seems no fine-tuning was done. But how did the authors train the learnable projection for dimensionality reduction (embedding head)?**
> Thank you for the question. To clarify, the learnable projection head was only applied during our fine-tuning phase, not for zero-shot baseline. For the baseline, we used the off-the-shelf models with their original, untrained feature heads (2048-D for ResNet, 768-D for Swin-T). In contrast, for our proposed fine-tuning method, we introduced a new learnable projection head (a fully connected layer) to both backbones, which was then trained on AutoFish dataset. This head maps features to a unified 512-D space, which is a standard metric learning procedure. This ensures fair performance comparison based purely on the quality of the backbone's features, rather than the arbitrary size of their original output dimensions. Figure 3 implicitly shows this modification, and we will explicitly clarify this distinction in the revised manuscript.
> 7. **The default image transformation is not clearly described.**
> We agree with the reviewer that there is a potential for confusion with the wording in the manuscript to represent the input image transformations. Figure 4 states the word "default", but it represents the "standard" input image transformations. We did not clearly specify and describe the standard image transformations given by PyTorch. We apologize for not detailing the configuration, as this custom pipeline was essential for achieving high Re-ID performance on our specific dataset. The standard transformations referenced (Random sized crop, center crop), optimized for large-scale classification, can potentially destroy fine-grained identity features. Our custom pipeline was utilized to preserve these features and obtain higher performance.
> 8. **The authors tested only Euclidean distance for retrieval; evaluating other distance metrics or similarity measures would be useful.**
> We appreciate the suggestion to evaluate other distance metrics. However, experimenting with more distance metrics would be hard when summarizing all the experiments. We also believe that this approach helps readers better comprehend the findings of our work. Our pipeline uses a learnable projection head that outputs L2-normalised embeddings. When comparing two-unit length vectors, the L2 (Euclidean) distance is mathematically and computationally efficient in measuring their angular separation. This measurement is directly equivalent to maximizing cosine similarity, which is one of the widely used and standard similarity measures for Re-ID.
> 9. **In Table 3, the authors did not explain why Separate Flipped (Q) / Touched Initial (G) gives better results than Touched Initial (Q) / Touched Initial (G), and similarly for the other combinations (Separate Initial (Q) / Touched Flipped (G) and Touched Flipped (Q) / Touched Flipped (G)).**
> We accept this point and will add this discussion to the manuscript if it gets accepted. The superior performance of a "Separated" (Unoccluded) query stems from the quality of its feature vector. A clear, unoccluded query image allows the model to extract a pure, highly discriminative feature vector. This robust vector is strong enough to find the correct match even when the gallery is "Touched" (Occluded). Conversely, a "Touched" query provides a low-quality, ambiguous input. This results in a suboptimal and less discriminative feature vector. When this "noisy" query vector is used to search an equally  "noisy" occluded gallery, the matching challenge is compounded, leading to significantly lower R1 and mAP@k values. In short, the results indicate that occlusion in the query is a greater deterrent to Re-ID performance than occlusion in the gallery. The model is far more robust when matching a clean query against a noisy gallery than when matching a noisy query against a noisy gallery.
> 10. **In the t-SNE plots, the authors did not explain why only 8 fish IDs were shown, nor how these IDs were selected.**
> The reason to show only 8 fish IDs is mentioned in Figure 7 of the manuscript. They were selected randomly to show the cluster distribution. Adding all 94 fish IDs in the test set is not feasible for visualizing the proper cluster distribution, and because they are 3D plots, they cannot be rotated to better visualize the cluster arrangement in the PDF file. We have been maintaining a GitHub repository (currently not available to the public) for sharing source codes and 3D interactive plots.
> Notably, we have conducted UMAP (Uniform Manifold Approximation and Projection) on the test data embeddings, and it has shown even clearer cluster arrangements, both species-wise and individual-wise. We will refer to our GitHub repository for all the source codes and 3D visualizations if the paper is accepted.

---

### Meta-Review · Area_Chair_pHVk · 2025-11-03

**Recommendation:** Accept (Poster)
**Confidence:** 4

**Metareview:**

All reviewers agree about the value of the paper, that presents a competent and well-executed application of state-of-the-art deep metric learning to an important domain (fisheries monitoring). While it lacks strong algorithmic novelty, it compensates with methodological rigor and empirical strength.  The rebuttal adequately addresses reviewers' concerns. I'm recommending acceptance as poster.

---

### Decision · Program_Chairs · 2025-11-05

**Decision:**

Accept (Poster)

**Comment:**

We recommend a poster presentation given the AC and reviewers recommendations.